# HPLC Analysis of the Urinary Iodine Concentration in Pregnant Women

**DOI:** 10.3390/molecules26226797

**Published:** 2021-11-10

**Authors:** Aniceta A. Mikulska, Dorota Filipowicz, Franciszek K. Główka, Ewelina Szczepanek-Parulska, Marek Ruchała, Michał Bartecki, Marta Karaźniewicz-Łada

**Affiliations:** 1Department of Physical Pharmacy and Pharmacokinetics, Poznan University of Medical Sciences, 60-781 Poznań, Poland; amikulska@ump.edu.pl (A.A.M.); glowka@ump.edu.pl (F.K.G.); 2Department of Endocrinology, Metabolism and Internal Medicine, Poznan University of Medical Sciences, 60-356 Poznań, Poland; dorota.filipowicz123@gmail.com (D.F.); ewelina@ump.edu.pl (E.S.-P.); mruchala@ump.edu.pl (M.R.); 3Department of Pediatric Cardiology, Poznan University of Medical Science, 60-572 Poznań, Poland; mbartecki@ump.edu.pl

**Keywords:** ion-pair HPLC–UV method, validation, in vivo application, iodine supplementation, pregnancy

## Abstract

Iodine is an essential component for fetal neurodevelopment and maternal thyroid function. Urine iodine is the most widely used indicator of iodine status. In this study, a novel validated ion-pair HPLC–UV method was developed to measure iodine concentration in clinical samples. A sodium thiosulfate solution was added to the urine sample to convert the total free iodine to iodide. Chromatographic separation was achieved in a Pursuit XRs C8 column. The mobile phase consisted of acetonitrile and a water phase containing 18-crown-6-ether, octylamine and sodium dihydrogen phosphate. Validation parameters, such as accuracy, precision, limits of detection and quantification, linearity and stability, were determined. Urinary samples from pregnant women were used to complete the validation and confirm the method’s applicability. In the studied population of 93 pregnant women, the median UIC was lower in the group without iodine supplementation (117 µg/L, confidence interval (%CI): 95; 138) than in the supplement group (133 µg/L, %CI: 109; 157). In conclusion, the newly established ion-pair HPLC–UV method was adequately precise, accurate and fulfilled validation the criteria for analyzing compounds in biological fluids. The method is less complicated and expensive than other frequently used assays and permits the identification of the iodine-deficient subjects.

## 1. Introduction

Iodine and selenium are crucial microelements for the proper functioning of the thyroid gland, including the synthesis of triiodothyronine (T3) and thyroxine (T4). They are closely related to fetal neurodevelopment, growth and basic metabolism [1,2,3,4]. Iodine deficiency (ID) is considered one of the major public health problems worldwide [3,5]. The primary strategy for the elimination of ID is universal salt iodization. Iodine deficiency prevention with global salt iodization programs began in the 20th century. Currently, it is recommended to limit salt intake, which is the main source of iodine, owing to the increased incidence of hypertension and other cardiovascular diseases [6,7,8]. Consequently, the consumption of various iodine-rich products, including seafood, fish, dairy products, eggs and meat, should be increased [9]. Nowadays, after a successful salt iodization program in Poland, the country is considered iodine-sufficient, but these analyses are not relevant to pregnant and lactating females [10]. During pregnancy, the thyroid hormones T4 and T3 production increase by almost 50%, which requires a higher iodine supply. As a result, salt iodization and iodine intake are insufficient [11,12], and additional supplementation is recommended in pregnant women [13]. Women who take an adequate dose of iodine before and during pregnancy have enough iodine stores and have no difficulty in adapting to the increased demand for thyroid hormones; accordingly, iodine levels remain stable throughout this period. In areas with mild-to-moderate iodine deficiency, iodine stores decline gradually from the first to the third trimester of pregnancy [9].

Iodine deficiency disorders depend on the severity and duration of dietary iodine deficiency at certain stages of life, particularly during fetal and infant development [2,14]. ID is the world’s leading cause of mental retardation in children and may impair the normal growth and development of the child, significantly affecting neurogenesis [9,14,15,16]. Severe ID during pregnancy (urinary iodine concentration (UIC) < 50 μg/L) is associated with many irreversible adverse effects, including disturbed nerve myelination, central nervous system (CNS) damage and cretinism, as well as an increased risk of miscarriage and premature birth. It may also lead to hypothyroidism in pregnant women and the fetus. Mild-to-moderate iodine deficiency (UIC of 50–149 μg/L) may also be associated with impaired psychomotor development, attention deficit hyperactivity disorders (ADHD) [11], decreased Intelligence Quotient (IQ) [17,18,19] and impaired cognitive outcomes [18,20]. Iodine supplementation during the pre-contraceptive period and in early pregnancy may reduce the risk of these disorders [13,15].

During pregnancy, the requirement for iodine increases [13]. The daily iodine intake recommended by the World Health Organization (WHO) for pregnant women is 250 μg/day. To achieve the proper daily intake of iodine, it is necessary to consume iodized table salt and additional supplementation of iodine (150–200 μg iodine) in the form of potassium iodide (KI) or multivitamin supplements containing KI [21]. The American Thyroid Association (ATA) recommends additional dietary daily iodine supplementation of 150 μg in the form of KI for pregnant women, which should be started three months in advance of planned pregnancy [9]. 

The effectiveness of iodine prophylaxis in pregnant women should be monitored [22,23]. The most common biomarkers for analyzing iodine status are UIC, serum thyroid-stimulating hormone, thyroglobulin levels, and thyroid volume [2,13,24,25,26]. More than 90% of absorbed iodine is excreted through the kidneys, so the UIC is the most widely used indicator of iodine status at the population level [26,27]. There are three methods for reporting the iodine status value from spot urinary collection. These methods include simple UIC, iodine-to-creatinine ratio (I/Cr) and estimated age/sex-adjusted 24 h iodine excretion [25,28]. Measurement of a spot UIC is commonly used and recommended by the WHO as a good tool for assessing iodine status in a population [22,29]. 

Table 1 shows the WHO reference values of median UIC for the classification of iodine concentration in pregnant women [23]. According to the WHO guidelines, the median UIC for pregnant women between 150 and 249 μg/L is consistent with adequate iodine intake [9,22,23].

Several analytical methods are available for measuring UIC, but only a few are routinely used. Most of them require a time-consuming sample preparation step or costly detection systems. Currently, inductively coupled plasma mass spectrometry (ICP–MS), characterized by high specificity and sensitivity, is considered the method of choice to provide a reliable measurement of UIC. However, ICP–MS equipment is not easily available in all laboratories, and requires qualified staff to operate it [30,31,32,33,34]. The other commonly used technique for iodine measurement is the time-consuming spectrophotometric method based on the Sandell–Kolthoff (S–K) reaction. In this procedure, iodide catalyzes the reduction of yellow-colored tetra-ammonium cerium (IV) sulfate to the colorless cerous form by arsenite, allowing the spectrophotometric detection of the color disappearance. The S–K method requires complex sample preparation, including digestion at high temperature and the use of significantly hazardous chemicals such as arsenic and cerium. Moreover, urine contains substances that can interfere with the compounds of the S–K reaction [31]. Most studies indicate that the above methods are comparable in terms of precision and accuracy, especially when the UIC is less than 300 μg/L, which covers the normal adult UIC range [31,34,35,36]. Up to now, only one high-performance liquid chromatography with ultraviolet detection (HPLC–UV) method has been described to determine iodine in biological fluids. The method applied simple protein precipitation with acetonitrile and a mobile phase with ion-pairing reagents to analyze iodine in rabbit plasma [37]. The ion-pair HPLC–UV method is a widely used analytical method. This technique is applied to separate compounds, including iodide, that contain ionizable or strongly polar groups that lead to poor retention of these compounds on the reversed-phase column [38]. 

Considering the importance of iodine deficiency detection and increased demand for iodine status determination, it is clear that a simple analytical method that could be easily available is needed. Therefore, this study aimed to develop and validate a simple ion-pair HPLC–UV method to determine iodine urine concentration in clinical samples and apply the method to assess iodine supplies in pregnant women.

## 2. Results

### 2.1. Development of the Ion-Pair HPLC–UV Method 

In the developed conditions, the retention time of I^−^ was about 15.5 min. The representative chromatograms of the blank sample, standard samples (at concentrations equal to limit of detection (LOD), limit of quantitation (LOQ) and 200 µg/L), and a urine sample of the pregnant woman are displayed in Figure 1. 

### 2.2. Validation Results

#### 2.2.1. Linearity, LOD and LOQ

The linearity of the method was evaluated at seven concentration levels. The resulting peak areas were processed and calibration curves were generated by Microsoft Excel. The standard calibration curves were linear over the concentration range of 50–300 μg/L, covering the required range for the analysis of iodine. The average equation of the calibration curve was: y = 66.6x. The correlation coefficient (r) for standard curves was 0.9994, indicating the method linearity. To check the significance of the value of b, Student’s *t*-test was performed. Under the stated experimental conditions, LOQ for the iodine analysis was 50 μg/L and LOD was 18 μg/L, which were sensitive enough for clinical utility.

#### 2.2.2. Accuracy and Precision

The validation results of the inter-day and intra-day accuracy and precision shown in Table 2 were within the European Medicines Agency (EMA) guidelines [39]. Intra-day and inter-day precision expressed by the relative standard deviation (%RSD) were less than 10%. Accuracy determined by relative error (%RE) was below 10%, except for the LOQ, which was 20%.

#### 2.2.3. Stability

For the stability evaluation of iodine in urine samples, the differences were calculated between the initial concentration and the concentration found after defined storage conditions. All analyses were performed for patient samples containing iodine at a low, medium and high level. The samples were prepared according to the procedure described in Section 4.4. The deviations expressed as %RE were within acceptable limits of ±15%. The results of stability assessment of iodine under various conditions, including long-term, short-term, autosampler and freeze-thaw stability, are illustrated in Table 3.

#### 2.2.4. In Vivo Study

The applicability of the validated ion-pair HPLC–UV method was confirmed in the analysis of iodine in urinary samples from 93 pregnant women. Figure 2 shows the measured concentrations of iodine in urine in all pregnant women. The recommended range of UIC in pregnant women by WHO is marked with green lines.

The median UIC in all studied women was 127 µg/L (%CI: 110; 145). The median UIC in patients with iodine supplementation (133 µg/L, %CI: 109; 157) was higher than in those who were not taking iodine-containing supplements (117 µg/L, %CI: 95; 138), but the difference was not statistically significant (*p* = 0.97). The distribution of ioduria is shown in Figure 3.

## 3. Discussion

The measurements of urinary iodine concentration in populations from selected regions help verify the iodine supplementation dose required to reach the recommended iodine supplies in pregnant women. Owing to the pivotal role of iodine for human health and the increasing need for iodine status assessment, an effective, cheap and uncomplicated urinary iodine detection method is required. Different techniques for UIC determination, such as S–K or ICP–MS, have been developed over the years. These methods require expensive instrumentation with qualified personnel (ICP–MS) or an initial digestion step to remove potential interfering substances before analysis by a kinetic colorimetric method [30]. Our study aimed to develop a simple and inexpensive HPLC–UV method to determine the urinary iodine concentration and confirm the applicability of this method in assessing iodine supplies in pregnant women. The sample preparation and the mobile phase composition were based on the method by Cui et al. [37] developed to determine iodine in rabbit plasma. The modifications needed to separate iodide from compounds originating from urine included a different chromatographic column, gradient elution of a mobile phase with slightly different pH and lower amount of ACN for protein precipitation. Sodium thiosulfate solution was added to urine samples to convert the total free iodine to iodide. The use of octylamine and 18-crown-6 ether in the water phase improved the resolution of iodine from endogenous components present in the urine samples. The –NH^3+^–I^−^ ion pair attenuated the negativity of iodide and increased its hydrophobicity, leading to the longer iodide retention on the C8 column. This effect was enhanced by the addition of 18-crown-6 ether, which forms stable complexes with I^-^ [37]. 

The developed HPLC–UV method was validated. LOD for the analysis of iodine was 18 μg/L, which is sensitive enough for clinical utility. The LOQ was 50 μg/L and was sufficient to identify the iodine deficiency. The current LOQ and LOD values were higher than previously reported by the highly specific ICP–MS and S–K methods (Table 4). The precision obtained in our study was <10% (Table 2). In contrast to the S–K method, the developed ion-pair HPLC–UV method does not require digestion or oxidation of the samples before the analysis. In addition, it does not use hazardous chemical. Compared to the ICP–MS method, our method is inexpensive and does not require specialized equipment with qualified staff [30,34]. Moreover, according to our knowledge, we present the first study on the iodine stability in urine during sample pretreatment and storage. Iodine proved to be stable in urine samples after three freeze-thaw cycles, in a long-term stability study (−20 °C for 73 days), in a short-term stability test (25 °C for 3 h) and in the autosampler test (24 h at 25 °C). Our newly developed method fulfils the validation requirements for the analysis of compounds in biological fluids. 

The method’s applicability was evaluated in the assessment of iodine status in pregnant women. The study population was not homogeneous, as it involved healthy women and those with mild thyroid disorders. However, we believe that the population is appropriate to confirm the usefulness of the HPLC–UV method in identifying iodine-deficient subjects. The WHO guidelines define iodine status for all pregnant women, regardless of concomitant diseases, as adequate when the median UIC is 150–249 µg/L and insufficient when <150 µg/L [23]. Moreover, in the studied population, the differences in the urine iodine concentration between healthy pregnant women and those with thyroid disorders were statistically insignificant (*p* = 0.208, data not shown). 

In our study, the pregnant women without and also with iodine supplementation displayed insufficient iodine intake. More than three-quarters of pregnant women without supplementation were iodine-deficient. Additionally, the study group was deficient in iodine, despite taking daily iodine doses offered as dietary supplements dedicated for pregnant females. The median UIC of all pregnant women (127 µg/L) was lower than sufficient iodine status according to WHO criteria. Our results are consistent with previous data. In recent years, several studies have assessed UIC in pregnant women. Most of them showed low median UIC, suggesting insufficient iodine status in pregnancy [25,41,42,43,44,45,46,47,48].

Despite the introduction of salt iodization programs, iodine deficiency remains a significant public health problem in Europe [12,49,50]. Further research should be performed on a larger group to confirm our results and develop optimal strategies for preventing and treating iodine deficiency.

### Limitations and Strength of the Study 

This study has some limitations. The first is the small number of individuals. It is worth considering expanding the study population, and the groups should be more homogeneous, with normal thyroid function and without levothyroxine treatment. Moreover, the study contained only Caucasian pregnant females from one province, although conducted at a tertiary reference center for gynecology, obstetrics and neonatology, currently the biggest gynecological hospital in Greater Poland. Another limitation is that women were recruited, and spot urine samples were collected, only during the third trimester of pregnancy. It is worth analyzing the iodine status, including all trimesters of gestation, using 24 h iodine collection samples or the iodine-to-creatinine ratio instead of spot urinary collection. Moreover, an internal standard (IS) was not used for the method validation.

The key strength of the study included the development and validation of a new ion-pair HPLC–UV method for the determination of iodine in urine. This is the first study to establish iodine status by the ion-pair HPLC–UV method in human urine to the best of our knowledge. The method is less complicated and expensive than other methods frequently used. Moreover, the stability study of iodine in urine samples was performed for the first time.

## 4. Materials and Methods

### 4.1. Chemicals and Reagents

Potassium iodide (99% purity), octylamine (99% purity), 18-crown-6 ether (99% purity), sodium thiosulfate (99% purity), phosphoric acid (BioUltra, ≥85% purity) were obtained from Sigma-Aldrich (Steinheim, Germany). Acetonitrile (HPLC-grade) was purchased from Merck KGaA (Darmstadt, Germany). Ultra-pure water was generated in-house using a Simplicity^®^ Water Purification System from Merck Millipore (Burlington, MA, USA).

### 4.2. Ion-Pair HPLC–UV Conditions and Apparatus

The study was conducted using an HPLC Agilent 1100 Series set (Agilent Technologies Inc., Santa Clara, CA, USA). To process the data, the Agilent ChemStation for LC 3D system (Agilent Technologies Inc., Waldbronn, Germany) was used. The chromatographic separation was performed on a Pursuit XRs C8 column (250 × 4.0 mm, 5 μm; Agilent Technologies Inc., Santa Clara, CA, USA). The column temperature was maintained constantly at 20 °C using a thermostatically controlled column oven. The mobile phase consisted of a mixture of water phase A (10 mmol/L 18-crown-6 ether, 5 mmol/L octylamine and 5 mmol/L sodium dihydrogen phosphate, pH adjusted to 5.95 with phosphoric acid) and acetonitrile (mobile phase B). Following its preparation, the water phase mixture was filtered under vacuum through a 0.45 μm membrane filter and degassed in the ultrasonic bath before use. Chromatographic separation of iodine from endogenous compounds was accomplished at a 1.2 mL/min flow rate with 12% mobile phase B. The injected sample volume was 50 μL. The UV detection was performed at 225 nm.

### 4.3. Preparation of Standard Solutions and Quality Control Samples

A stock solution of iodine at a concentration of 1 mg/mL was prepared by dissolving 13.08 mg of KI in 10 mL of ultra-pure water. Working standard solutions at 0, 500, 750, 1000, 1500, 2000, 2500 and 3000 μg/L of iodine were prepared from stock solution by dilution in ultra-pure water. 

Calibration standards were prepared by spiking 180 μL of ultra-pure water with 20 μL of the working standard solutions to give nominal concentrations of 0, 50, 75, 100, 150, 200, 250 and 300 μg/L. The calibration curve standards were freshly prepared from the working standard solutions for each validation and assay run. Quality control samples (QCs) were independently prepared directly before analysis at four concentration levels of 50, 75, 150 and 250 μg/L. 

### 4.4. Sample Preparation

A volume of 200 μL of sodium thiosulfate solution (5 g/L) was added to 200 μL calibration standards or QCs. Tubes were capped and then mixed by vortex for 1 min. After that, 50 μL of acetonitrile was added to each sample, followed by vortexing for 2 min. The mixture was centrifuged at 2500× *g* for 10 min. After mixing, the sample was conveyed to a new glass vial, and 50 μL was injected into the HPLC system. Patients’ urine samples (200 μL) were also subjected to the above procedure. For each subject sample, the analysis was performed in duplicate. 

### 4.5. Method Validation

The developed method was validated for linearity, precision, accuracy, stability in different conditions, LOD and LOQ following guidelines on bioanalytical method validation of the EMA [39].

Linearity was estimated for the peak area of iodide as a function of the iodine concentration covering the range of 50–300 μg/L in urine. Equations generated from the calibration curve were used for calculating the concentrations of iodine in the patient’s urine. 

The method inter-day and intra-day precision and accuracy were calculated based on QCs. Five replicates were spiked with LOQ (50 μg/L), low (75 μg/L), medium (150 μg/L) and high (250 μg/L) concentrations. The analysis of the total number of replicates was conducted during one day and over five consecutive days. The precision was calculated as relative standard deviation (RSD; SD/C_measured_)·100%. The accuracy was estimated as a relative error (%RE; ((C_nominal_ − C_measured_) × 100%/C_nominal_).

LOD was defined as the smallest concentration of iodine that could be detected with the corresponding signal-to-noise (S/N) ratio greater than 3:1. LOQ was determined as the lowest concentration of iodine determined by the method with the precision and accuracy ≤20%. 

Iodine stability was evaluated with three concentrations (low, medium and high) of the patient’s urine in three replicates for each concentration after three freeze-thaw cycles, long-term (over 2 months at −20 °C) and short-term (3 h at 25 °C) storage, and after standing of the prepared samples in autosampler for 24 h at 25 °C. The samples were prepared according to the procedure described above. Based on the EMA guidelines for bioanalytical method validation, the stability of the analyte is confirmed if the deviation from the nominal concentration is within ±15% [39]. For the nominal concentration, we considered the iodine concentration measured after the first thaw of the urine sample. 

### 4.6. In Vivo Application 

Following the validation of the ion-pair HPLC–UV method, it was used to determine the iodine concentration in the urine of 93 Caucasian pregnant women. Patients were recruited at the Department of Obstetrics and Gynecology of Poznan University of Medical Sciences at planned admission to the obstetric ward of the gynecological hospital. They were divided into two groups: 61 women declaring iodine supplementation (study group) to the amount of 150–200 μg per day (183 ± 26 µg/daily) and 32 without iodine supplementation (control group). Both groups consisted of pregnant women of similar age (32.7 ± 3.8 years) who lived in the same geographical area of Greater Poland. Women who had poor health (including kidney or liver dysfunction, the necessity of chronic medication intake other than levothyroxine) according to physical examination and laboratory analyses following an interview were excluded from the study. Included females were healthy or affected by benign thyroid disorders (thyroid nodules, autoimmune thyroid diseases, hypothyroidism) with or without levothyroxine treatment. A single spot urine sample was obtained from mothers at admission before delivery to measure iodine concentration. The urine samples were stored at −20 °C until use and were allowed to thaw at room temperature before processing.

The study was approved by the Local Bioethics Committee of Poznan University of Medical Sciences (approval number 104/19). Participation in the study was voluntary. Each participant gave written informed consent after having been informed about the project’s purpose and course. The study has been conducted according to outlined ethical principles in the Declaration of Helsinki [51]. 

### 4.7. Data Processing and Statistical Analysis

Statistical analysis of the results was conducted using Statistica 13 software with Medical Set (StatSoft, Tulsa, OK, USA). Continuous data were presented as median and confidence interval (%CI). The normality of the data distribution was assessed using the Shapiro–Wilk test. The results were analyzed statistically, using elements of descriptive statistics and statistical procedures, such as correlation analysis (Spearman test for non-normal distributions). Comparisons between groups were performed using the Mann–Whitney U test for non-normal data distribution. The level of statistical significance was taken as *p* < 0.05.

## 5. Conclusions

This developed simple and low-cost ion-pair HPLC–UV method for determining iodine in human urine is accurate, repeatable, reproducible, and precise and was applied to analyze iodine in the urine of 93 pregnant women. Most pregnant females were iodine-deficient, according to WHO guidelines, independent of iodine supplementation. Further research, with a larger and more homogeneous population of pregnant women, should be performed to evaluate urinary iodine levels in this population. This newly established and validated method is less complicated and expensive than other frequently used assays and may be applicable for monitoring iodine status to prevent iodine deficiency, especially in pregnant women.

## Figures and Tables

**Figure 1 molecules-26-06797-f001:**
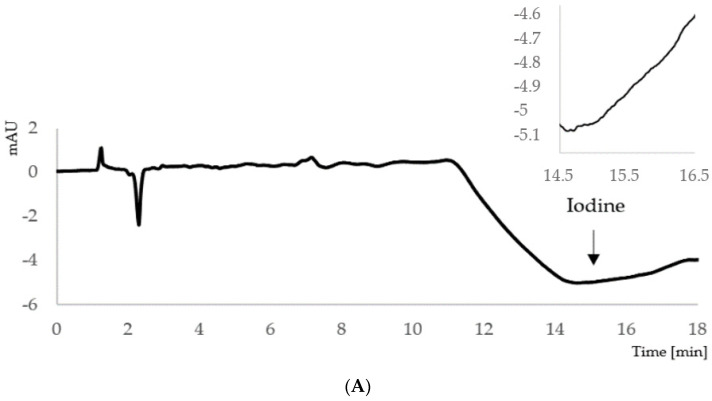
Representative chromatograms of (**A**) blank sample; (**B**) limit of detection (LOD); (**C**) limit of quantitation (LOQ); (**D**) calibration standard at concentration 200 µg/L; (**E**) urine sample of the pregnant woman (determined iodine concentration 271 µg/L). HPLC conditions: Pursuit XRs C8 column (250 × 4.0 mm, 5 μm); mobile phase: mixture of water phase (18–crown–6 ether, octylamine, sodium dihydrogen phosphate; pH 5.95) and acetonitrile (88:12, *v*/*v*); UV detection at 225 nm.

**Figure 2 molecules-26-06797-f002:**
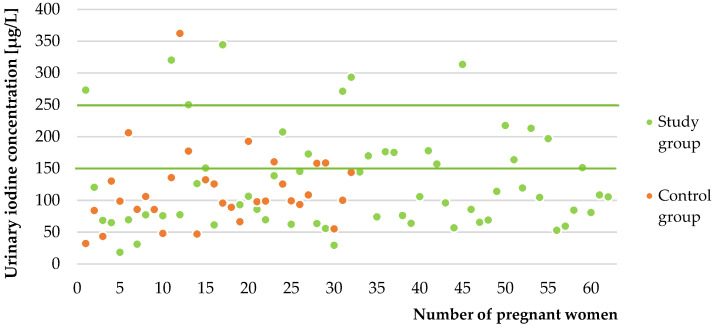
Urinary iodine concentrations in all pregnant women.

**Figure 3 molecules-26-06797-f003:**
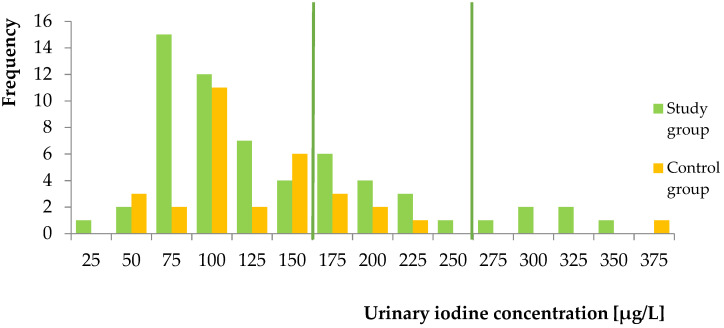
Histogram of the number of women with different levels of urinary iodine concentration.

**Table 1 molecules-26-06797-t001:** Median UIC categories in pregnant women according to the WHO [23].

Category of Iodine Supply in Pregnant Women	Median Urinary Iodine Concentration [μg/L]
Insufficient	<150
Adequate	150–249
Above requirements	250–499
Excessive	≥500

**Table 2 molecules-26-06797-t002:** Intra-day and inter-day precision and accuracy.

QCs Concentration	Precision (RSD, %)		Accuracy (%RE)	
	Within-Run *n* = 5	Between-Run *n* = 5	Within-Run *n* = 5	Between-Run *n* = 5
LOQ (50 μg/L)	1.07	9.72	20.00	9.43
Low (75 μg/L)	8.44	9.12	0.06	2.16
Medium (150 μg/L)	8.57	9.71	4.88	4.77
High (250 μg/L)	2.92	4.69	0.38	0.99

QCs—quality control samples; LOQ—limit of quantitation, *n*—number of samples; %RSD—relative standard deviation; %RE—relative error.

**Table 3 molecules-26-06797-t003:** The stability of iodine in urine samples under different conditions.

Concentration	High (*n* = 3)	Medium (*n* = 3)	Low (*n* = 3)
**Long-term stability** (−20 °C for 2 months)			
Initial concentration (mean ± SD, µg/L)	273 ± 2	145 ± 15	68.2 ± 6.0
Concentration determined after stability study (mean ± SD, µg/L)	263 ± 10	154 ± 6	73.3 ± 2.5
Accuracy (RE, %)	3.79	5.77	7.54
**Short-term stability** (25 °C for 3 h)			
Initial concentration (mean ± SD, µg/L)	278 ± 22	196 ± 9	96.2 ± 7.9
Concentration determined after stability study (mean ± SD, µg/L)	263 ± 21	188 ± 8	108 ± 10
Accuracy (RE, %)	5.24	3.72	12.43
**Freeze-thaw stability** (After three freeze-thaw cycles)			
Initial concentration (mean ± SD, µg/L)	273 ± 2	145 ± 15	68.2 ± 6.0
Concentration determined after stability study (mean ± SD, µg/L)	260 ± 18	127 ± 3	61.2 ± 4.6
Accuracy (RE, %)	4.76	12.75	10.27
**Stability in autosampler** (25 °C for 24 h)			
Initial concentration (mean ± SD, µg/L)	306 ± 6	164 ± 30	107 ± 3
Concentration determined after stability study (mean ± SD, µg/L)	307 ± 5	173 ± 9	117 ± 3
Accuracy (RE, %)	0.29	5.17	9.28

**Table 4 molecules-26-06797-t004:** Characteristics of the methods used for the determination of iodine in urine.

Method	Sample Preparation	Urine Volume	LOD [μg/L]	LOQ [μg/L]	Precision [%]	Ref.
Ion-pair HPLC–UV	200 μL of sodium thiosulfate solution and 50 μL of acetonitrile	200 μL	18	50	<10	-
ICP–MS	dilution (1:20) in an aqueous solution of 0.5 mL/L Triton X-100 and suprapure concentrated HCl	500 μL	4	20	n.a.	[36]
ICP–MS	dilution (1:9) in 0.1% EDTA and 0.1% ammonia	1 mL	3.3	10	2.2	[31]
ICP–MS	dilution (1:20) by adding an aqueous solution containing 0.1 mg/L NH4OH, 0.1 g/L EDTA, 5 mg/L n-butanol and 0.1% Triton X-100	200 μL	0.1	n.a.	n.a.	[40]
ICP–MS	dilution (1:9) using water with 1.5% isopropanol and 7 mmol hydrous ammonium	200 μL	0.87	8.1	<8	[34]
S–K method	ammonium persulfate as the digestion regent	n.a.	2.0	n.a.	<4	[34]
S–K method	dilution with water (1:1), perchloric acid digestion, incubation 25 min at 200 °C	200 μL	n.a.	13	<19.2	[31]

n.a.—not available; LOD—limit of detection, LOQ—limit of quantitation; ICP–MS—Inductively coupled plasma mass spectrometry; S–K method—Sandell–Kolthoff method.

## Data Availability

The data presented in this study are available on request from the corresponding author. The data are not publicly available due to privacy restrictions.

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
