# Peer review of "HPLC Analysis of the Urinary Iodine Concentration in Pregnant Women"

_molecules, 2021, doi:10.3390/molecules26226797_

Round 1

Reviewer 1 Report

The work is well conducted but presents important gaps:

the methodology of setting of your HPLC-UV method is valid. However, it would be appropriate and necessary to use method-comparison methodology to compare your results with a method already validated and in use.

From a clinical point of view the choice of the samples have too many biases because the sample size is too small and inappropriate (included subjects were healthy or affected by benign thyroid disorders in the same group).

Author Response

Comment no 1: The work is well conducted but presents important gaps:

the methodology of setting of your HPLC-UV method is valid. However, it would be appropriate and necessary to use method-comparison methodology to compare your results with a method already validated and in use.

Authors' response: Thank you for your careful study of the manuscript. The aim of our study was to develop and validate a new simple and low-cost method for the determination of iodine in urine. The next stage of our research may be to compare the new method with commonly used methods in order to verify the results.

Comment no 2: From a clinical point of view the choice of the samples have too many biases because the sample size is too small and inappropriate (included subjects were healthy or affected by benign thyroid disorders in the same group).

Authors' response: Thank you for your valuable comment. Urinary samples from pregnant women were used to complete the validation and to confirm the applicability of the method. The main aim of this study was to develop and validate a simple ion-pair HPLC-UV method for the determination of iodine urine concentration in clinical samples, and to apply the method to assess iodine supplies. The clinical aspect is only an additional element in our manuscript. However, in section 3.1. Limitations and strength of the study, we added the sentence: “It is worth considering expanding the study population and the groups should be more homogeneous, all with normal thyroid function and without levothyroxine treatment.” 

Reviewer 2 Report

The paper has  been modified deleting part of the discussion section. The authors should consider the non honogenous group they are analyzing and insert a discussion on this point.

Author Response

Comment no 1: The paper has been modified deleting part of the discussion section. The authors should consider the non honogenous group they are analyzing and insert a discussion on this point.

Authors' response: Thank you for your valuable comment. As suggested by the Reviewer, appropriate correction was made in the article. We added information about non-homogeneous group in section 3. Discussion. Moreover, in section 3.1. Limitations and strength of the study, we indicated that: “the study groups should be more homogeneous, all with normal thyroid function and without levothyroxine treatment”.

Reviewer 3 Report

The current version of the manuscript is an improvement over the original. A few comments still need to be taken into account.

  1. Table 4: Characteristics of the method proposed in this paper should be introduced for comparison.
  2. Figure 1: the corresponding experimental conditions should be indicated in the figure caption.

Author Response

Comment no 1: The current version of the manuscript is an improvement over the original. A few comments still need to be taken into account.

Table 4: Characteristics of the method proposed in this paper should be introduced for comparison.

Authors' response: Thank you for your careful review. In the new version of the manuscript, we have added in Table 4 the characteristics of our newly developed method to compare with other methods.  

 Comment no 2: Figure 1: the corresponding experimental conditions should be indicated in the figure caption.

Authors' response: As suggested by the Reviewer, we have added the experimental conditions in the Figure 1 caption.

Round 2

Reviewer 1 Report

In this version of the manuscript the Authors have received my suggestions and have modified the text appropriately. In my opinion the manuscript can now be accept in present form.

This manuscript is a resubmission of an earlier submission. The following is a list of the peer review reports and author responses from that submission.

Round 1

Reviewer 1 Report

The HPLC-UV detection of the urine I concentration should be given with a clear demonstration of the typical chromatograms including LOD, LOQ, typical reference stand, and subjects samples, all in full range and time scale with adequate expansion.

Figure 2 should be consistent with the In vivo study of the iodine concentration in urine of 93 pregnant women, with the levels and subjects numbers clearly indicated.

Author Response

We are grateful to the Reviewers for their critical comments. Based on these suggestions, we have made careful modifications to the original manuscript.   

Comment no 1.

The HPLC-UV detection of the urine I concentration should be given with a clear demonstration of the typical chromatograms including LOD, LOQ, typical reference stand, and subjects samples, all in full range and time scale with adequate expansion.

Response: Thank you for your valuable comments. In the new version of the manuscript, we have added the HPLC chromatograms, including LOD, LOQ, calibration standard at concentration 200 µg/L and patients samples in full range and time scale with adequate expansion.

  • Comment no 2.

Figure 2 should be consistent with the In vivo study of the iodine concentration in urine of 93 pregnant women, with the levels and subjects numbers clearly indicated.

Response: As suggested by the Reviewer, we added Figure 2, which contains clearly indicated levels of iodine in urine and the patient numbers.

Reviewer 2 Report

In this manuscript the Authors describe a novel validated ion-pair HPLC-UV method to measure iodine concentration in clinical samples. The conclude that this method allowed to identify a population of iodine-deficient pregnant females who require additional iodine supplementation.

Major Revisions:

-In line 109 you describe that "S – K method requires a long incubation time and digestion at high temperature", in reality with current equipment a long incubation is not necessary; see PMID: 23462940.

-The authors report that their method is accurate and precise, but they should validate the results with the other commonly used technique for measuring iodine.

-The group of pregnant women used is too small from a statistical point of view. Moreover, it is not a homogeneous group because authors enrolled healthy subjects and with thyroid disease in the same group. Finally, talking about iodine deficiency, a 24-hour urine collection sample should be analyzed instead of urine spot samples.

Minor Revision:

In fig.1 it would be better to use the same scale for all 3 graphs.

Author Response

We are grateful to the Reviewers for their critical comments. Based on these suggestions, we have made careful modifications to the original manuscript. 

Comment no 1.

Major Revisions:

In line 109 you describe that "S – K method requires a long incubation time and digestion at high temperature", in reality with current equipment a long incubation is not necessary; see PMID: 23462940.

Response: Thank you for your valuable comments. Appropriate corrections were made in the article.

We changed “S – K method requires a long incubation time and digestion at high temperature” into “S – K method requires complex sample preparation, including digestion at high temperature and use of significant chemical hazards such as arsenic and cerium. Moreover, urine contains substances which can interfere with compounds of the S-K reaction”.

  • Comment no 2.

The authors report that their method is accurate and precise, but they should validate the results with the other commonly used technique for measuring iodine.

Response: Thank you for your valuable suggestion. The aim of our study was to develop and validate a new simple and low-cost method for the determination of iodine in urine. In the next stage of our research, we are going to compare the new method with commonly used methods to verify the results.

  • Comment no 3.

The group of pregnant women used is too small from a statistical point of view. Moreover, it is not a homogeneous group because authors enrolled healthy subjects and with thyroid disease in the same group.

Response: Thank you for your pertinent review. Another manuscript is in preparation for comparing urinary iodine concentration, thyroid function and several other biochemical parameters in healthy subjects and those with thyroid disease.

  • Comment no 4.

Finally, talking about iodine deficiency, a 24-hour urine collection sample should be analyzed instead of urine spot samples.

Response: Thank you for your valuable note. According to WHO recommendations [WHO. Urinary Iodine Concentrations for Determining Iodine Status in Populations, 2013], iodine concentrations measured in urine samples collected in the morning (or from other spot urine collections) have been shown to adequately assess a population’s iodine status. 24-hour sampling is harder to achieve, but not necessary. Our studied pregnant women came to the hospital only for a short follow-up visit, therefore the collection of 24-hour urine collection samples would be difficult.

  • Comment no 4.

Minor Revision:

In fig.1 it would be better to use the same scale for all 3 graphs.

Response: According to the Reviewer’s comment, in Fig. 1 the same scale was used for all HPLC chromatograms. Moreover, we added the chromatograms of LOD and LOQ in full range and time scale.

Reviewer 3 Report

The authors present an ion-pair HPLC-UV method to measure iodine concentration in clinical samples focusing on pregnant women. As the authors say the number of clinical samples is limited and also women affected by benign thyroid disorders (thyroid nodules, autoimmune thyroid diseases, hypothyroidism) with or without levothyroxine treatment are included. The authors should consider separately women with thyroid disorders compared to healthy. 

The main issue is related to the validation. The method here proposed reports LOD and LOQ higher compared to ICP or S-K and the authors should be sure that their results are in agreement with reported methods. In my opinion, the same samples should be analyzed by using ICP or S-K protocols.

The authors list in paragraph 3 a lot of results from literature that make annoying the reading. The results can be resumed in one table and a more critical discussion should be added instead.

Many errors in decimal digits are present in table 3. for instance, 273.11 ± 1.76 should be 273 ± 2 and so.

Author Response

We are grateful to the Reviewers for their critical comments. Based on these suggestions, we have made careful modifications to the original manuscript.  

Comment no 1.

The authors present an ion-pair HPLC-UV method to measure iodine concentration in clinical samples focusing on pregnant women. As the authors say the number of clinical samples is limited and also women affected by benign thyroid disorders (thyroid nodules, autoimmune thyroid diseases, hypothyroidism) with or without levothyroxine treatment are included. The authors should consider separately women with thyroid disorders compared to healthy.

Response: Thank you for your valuable comments. Currently, we are preparing another article that will compare urinary iodine concentration, thyroid function and several other biochemical parameters in healthy pregnant women and those with thyroid disease.

  • Comment no 2.

The main issue is related to the validation. The method here proposed reports LOD and LOQ higher compared to ICP or S-K and the authors should be sure that their results are in agreement with reported methods. In my opinion, the same samples should be analyzed by using ICP or S-K protocols.

Response: Thank you for your careful study of the manuscript. The aim of our study was to develop and validate a new simple and low-cost method for the determination of iodine in urine. Usually, the variation in iodine status of the population ranges from 50 to 300 μg/L. In our method LOD was 18 μg/L, which is sensitive enough for clinical utility and LOQ was 50 μg/L, allowing the identification of iodine deficiency. In the future, the same samples can be analyzed using the ICP or S-K protocols, but at the present stage of the research it is not possible for us.

  • Comment no 3.

The authors list in paragraph 3 a lot of results from literature that make annoying the reading. The results can be resumed in one table and a more critical discussion should be added instead.

Response: According to the reviewer's comment, the results of other studies (most of the information from lines 217-234) in the discussion has been removed and are presented in Table 4.

  • Comment no 4.

Many errors in decimal digits are present in table 3. for instance, 273.11 ± 1.76 should be 273 ± 2 and so.

Response: As suggested by the reviewer, the decimal digits have been changed.

Reviewer 4 Report

The manuscript entitled "HPLC analysis of the urinary iodine concentration in pregnant women" developed a simple and low-cost ion pair HPLC-UV method for the determination of iodine in human urine by adding sodium thiosulfate solution to urine sample. The method is accurate, repeatable, reproducible, and precise and was successfully applied to analyze iodine in the urine of pregnant women. The following points may be considered.

  1. The introduction is too expatiator. The 2nd and 3th paragraphs should be deleted or simplified. More details related to ion pairs HPLC using for the determination of iodine could be added. The innovativeness of this work should be pointed out more clearly.
  2. In Fig 1, it is better to place the three curves in the same coordinate system.
  3. In Section 2.2.1, the corresponding calibration curve should be given.
  4. In line 154, not "section 2.4" but "section 4.4".
  5. In line 157-163, the corresponding data can be clearly seen in Table 3, so it is unnecessary to make too much written description.
  6. Table 3 listed as “Stability in autosampler (20 °C for 24h)”. But in line 386, the temperature is 25 °C. Which is correct?
  7. Line 206,ref 35 should be 37.
  8. Line 217-234, this paragraph should be deleted and the corresponding content should be listed as a table, which could make it clear for reading.
  9. Line 214, C18 column, and, Line 333, C8 column. Please check it.
  10. The WHOLE HPLC chromatograms obtained from actual urine samples should be added. It is important for an analysis work.
  11. Except for the proposed HPLC method, whether the same urine sample has been detected by other method? Whether the result is same?

Author Response

We are grateful to the Reviewers for their critical comments. Based on these suggestions, we have made careful modifications to the original manuscript.

Comment no 1.

The introduction is too expatiator. The 2nd and 3th paragraphs should be deleted or simplified. More details related to ion pairs HPLC using for the determination of iodine could be added. The innovativeness of this work should be pointed out more clearly.

Response: In accordance with the reviewer's comment, the introduction has been simplified (paragraphs 2 and 3 have been shortened). Information about the ion-pair HPLC method used for the determination of iodine was also added. Moreover, the innovativeness of this work was pointed out more clearly.

  • Comment no 2.

In Fig 1, it is better to place the three curves in the same coordinate system.

Response: As suggested by the reviewer in Fig. 1, the curves were presented in the same coordinate system.

  • Comment no 3.

In Section 2.2.1, the corresponding calibration curve should be given.

Response: We have added in the article: “The average equation of the calibration curve was: y = 66.6 x”.

  • Comment no 4.

In line 154, not "section 2.4" but "section 4.4".

Response: Thank you for your careful reading. We changed section 2.4 into section 4.4.

  • Comment no 5.

In line 157-163, the corresponding data can be clearly seen in Table 3, so it is unnecessary to make too much written description.

Response: Thank you for your valuable clue. In lines 157-163, the data for the stability under different conditions are removed and presented only in Table 3.

  • Comment no 6.

Table 3 listed as “Stability in autosampler (20°C for 24h)”. But in line 386, the temperature is 25 °C. Which is correct?

Response: Thank you very much for error detection. In Table 3, the temperature listed as "Stability in autosampler (20°C for 24h)" should be 25°C. Appropriate corrections have been made.

  • Comment no 7.

Line 206ref 35 should be 37.

Response: Thank you for your careful study of the manuscript and associated literature. In line 206, the reference should be 37, therefore an appropriate change has been made.

  • Comment no 8.

Line 217-234, this paragraph should be deleted and the corresponding content should be listed as a table, which could make it clear for reading.

Response: As suggested by the reviewer, most of the information from lines 217-234 has been removed and the content is presented in Table 4.

  • Comment no 9.

Line 214, C18 column, and, Line 333, C8 column. Please check it.

Response: As suggested by the reviewer, we checked the column used in the study. The chromatographic separation was achieved in a Pursuit XRs C8 column.

  • Comment no 10.

The WHOLE HPLC chromatograms obtained from actual urine samples should be added. It is important for an analysis work.

Response: Thank you for your valuable comment. In the new version of the manuscript, the full HPLC chromatograms, including LOD, LOQ, calibration standard at concentration 200 µg/L and a patient’s urine sample in full range and time scale have been added.

  • Comment no 11.

Except for the proposed HPLC method, whether the same urine sample has been detected by other method? Whether the result is same?

Response: Urine samples were determined only by the proposed ion-pair HPLC-UV method. They were not assessed by other methods. In the future, the same samples can be analyzed using the ICP or S-K protocols, but at the present stage of the research it is not possible for us.

Round 2

Reviewer 2 Report

In this revised version of the manuscript the Authors have only partially responded to my previous requests. The underlying problems of patient selection and sample size remain unsolved. The patient groups are not homogeneous, and the conclusions to which the authors lead cannot be accepted. In particular, in line 424 the sentence "Our results indicate that the iodine status of pregnant women was insufficient by WHO guidelines. Most pregnant females were iodine-deficient independently from iodine supplementation." it does not have the sufficient basis to be considered valid. As the Authors themselves say, at line 199 "The aim of our study was to develop a simple and inexpensive HPLC-UV method to determine the urinary iodine concentration". This has been pointed out and the Authors should thrill here, then concentrating on a dedicated work to the application of the method, which includes a large series of pregnant women, selected without thyroid disease (even benign), supplemented with homogeneous supplements to be able to be compared, and perhaps using spot urine, but normalized to urinary creatinine to compensate for not using the 24-hour collection. These are just a few suggestions.
For the above reasons, this manuscript as it stands cannot be accepted.

Reviewer 3 Report

The authors reply to my concerns even if the comparison amnog methods has been posticipated in another paper. Some minor english error should be checked.

Reviewer 4 Report

The contents of the 2nd and 3th paragraphs in introduction seems less relevant to the analysis work. The reviewer still suggests that the 2nd and 3th paragraphs should be deleted or shortened to one or two sentences.